# Object-Aware Audio-Visual Sound Generation

## Abstract

Generating accurate sounds for complex audio-visual scenes is challenging, especially when multiple objects and sound sources are present. In this paper, we introduce an *object-aware sound generation* model that aligns generated sounds with visual objects in a scene. By grounding sound generation in object-centric representations, our model learns to associate specific visual objects with their corresponding sounds. We fine-tune a conditional latent diffusion model with dot-product attention to improve sound-object alignment. At test time, users can compositionally generate sounds by selecting objects via segmentation masks. We theoretically validate our test-time object-grounding ability, ensuring that even subtle sounds can be represented. Quantitative and qualitative evaluations show that our model outperforms baselines, achieving better alignment between objects and their associated sounds.

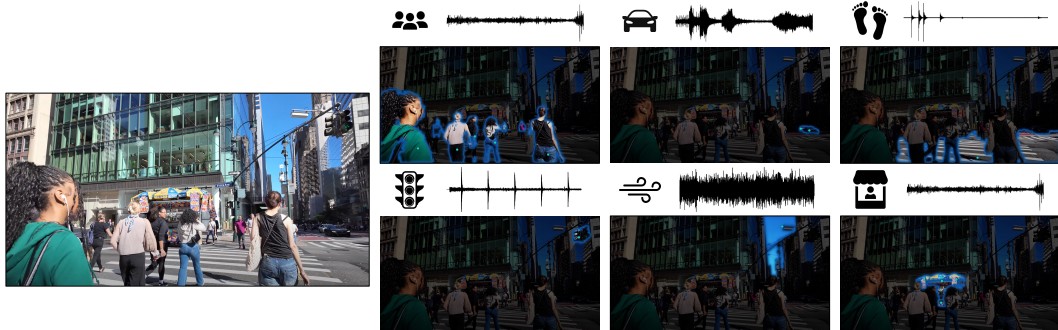

Input Image        Audio generated to match a user-selected object

Figure 1: **Object-aware sound generation**. We generate sound aligned with specific visual objects in complex scenes. Users can select objects in the scene using segmentation masks, and the model generates audio corresponding to the selected objects. Here, we show a busy street with multiple sound sources (left). After training, our model generates object-specific audio (right), such as crowd noise for people, engine sounds for cars, and ambient wind for the sky. **Please refer to our supplement and project webpage to watch and listen to the results.**

## 1 Introduction

Generating the full sound texture (McDermott & Simoncelli, 2011) of real-world environments is a significant challenge in audio and audio-visual research. While early models have focused on synthesizing sound based on scene categories, text descriptions, and visual contexts (Kong et al., 2019; Yang et al., 2023; Van Den Doel et al., 2001), they often fail to represent specific sound sources in complex environments. In scenes such as a busy city street (Figure 1), where multiple distinct sound events (e.g., car engines, footsteps, crowd noise) co-occur, these models often produce incomplete soundscapes (Pijanowski et al., 2011), overlooking important audio events.

Existing approaches can be largely classified as vision-based or text-based. Vision-based models (Sheffer & Adi, 2023) attempt to synthesize sound by analyzing the entire visual scene, but in environments with many overlapping sound sources, they tend to generate blended audio that misses subtle yet important details, like footsteps. Text-based models (Liu et al., 2023) respond to detailed prompts but face a similar challenge: certain sound events are either *forgotten* or *underrepresented* due to differences in the weight of each event in the latent space. For example, given a prompt de-

scribing both prominent and subtle sounds in a scene, the model might focus on only some of these events, omitting others like footsteps, even though they were explicitly mentioned (Wu et al., 2023). This occurs because the model assigns less importance to certain sounds, causing them to be ignored or poorly generated. While some have attempted to manually reweight sound events in the latent space (Xue et al., 2024), such interventions remain labor-intensive and impractical for large-scale applications.

To overcome these limitations, we propose an *object-aware sound generation* model that grounds sound generation in the visual domain. Inspired by object-centric learning (Greff et al., 2019), which decomposes scenes into discrete objects, our model associates visual objects with their corresponding sound sources, ensuring that no sound events are overlooked. We build on an off-the-shelf conditional audio generation model (Liu et al., 2023), enhancing it with dot-product attention (Vaswani et al., 2017) to learn sound-object associations through self-supervision. This method overcomes the problem of *forgetting* sound events, enabling the generation of various relevant sounds in complex scenes. To provide finer control and interactivity, we replace the attention with segmentation masks (Kirillov et al., 2023) at test time, allowing users to select specific objects in a scene (e.g., cars, groups of people) to generate the corresponding sounds within simple mouse clicks. This ensures that even subtle sound events, like footsteps or distant conversations, are captured accurately by grounding sound generation in specific objects rather than relying on scene-wide analysis.

Through quantitative evaluations and human perceptual studies, we demonstrate that our model outperforms existing baselines, generating more complete and contextually relevant soundscapes. In addition, we provide qualitative results and theoretical analysis demonstrating that our object-grounding mechanism is functionally equivalent to segmentation masks. Through our evaluations, we show:

- Visual grounding from text provides supervision for learning compositional sound generation.
- Specifying different objects within a scene leads to predictable changes in the types of generated sounds.
- Our model learns to generate sound from in-the-wild visual data.

## 2 RELATED WORK

**Predicting sound from images and text.** Generating sounds from visual and textual inputs has gained notable attention recently. Image-based methods focus on synthesizing sounds from visual cues such as physical interactions (Van Den Doel et al., 2001; Owens et al., 2016), human movements (Gan et al., 2020; Su et al., 2021; Ephrat & Peleg, 2017; Prajwal et al., 2020; Hu et al., 2021), musical instrument performances (Koepke et al., 2020), and content from open-domain images and videos (Zhou et al., 2018; Iashin & Rahtu, 2021; Sheffer & Adi, 2023; Luo et al., 2023). These approaches typically generate audio that corresponds to the entire visual scene without isolating individual sound sources, resulting in holistic sound generation. Text-based methods aim to produce sounds from textual descriptions using generative models like GANs and diffusion models (Yang et al., 2023; Kreuk et al., 2023; Liu et al., 2023; Huang et al., 2023b). However, when prompts contain multiple sound events, these methods often struggle to capture all the desired audio elements (Wu et al., 2023), potentially missing some sounds. Unlike these models, our method distinguishes itself by generating sounds compositionally and creating individual audio outputs for user-selected objects within images. This offers enhanced control and precision in sound generation.

**Object discovery.** Object-centric learning aims to represent visual scenes as compositions of discrete objects, enabling models to understand and manipulate individual entities within a scene. Unsupervised object discovery methods have been developed to decompose scenes into object representations without explicit annotations (Greff et al., 2019; Burgess et al., 2019). The Slot Attention mechanism (Locatello et al., 2020) introduced a way to learn such representations by utilizing a set of latent variables, or "slots," that iteratively attend to different parts of the input to capture individual objects. Subsequent works (Greff et al., 2019; Burgess et al., 2019) have sought to enhance the stability and robustness of these models. In the audio-visual realm, prior studies have explored object discovery (Arandjelovic & Zisserman, 2018; Rouditchenko et al., 2019; Afouras et al., 2020; Chen et al., 2021; Mo & Morgado, 2022; Hamilton et al., 2024) by leveraging the correspondence between audio and visual modalities. However, these methods primarily focus on recognition and localization tasks and do not address the generation of audio content based on visual inputs. In

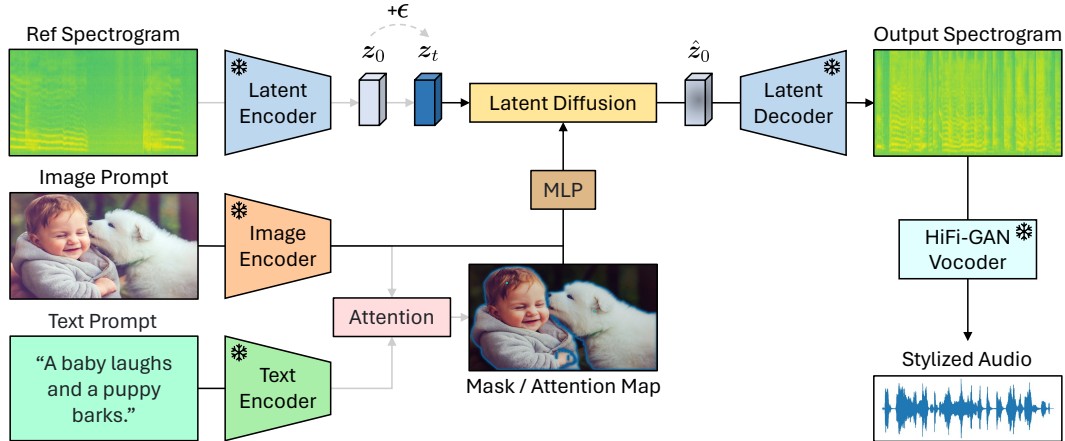

Figure 2: **Model architecture.** We encode the reference spectrogram via a pre-trained latent encoder. An image and text prompt are processed by separate encoders, and their embeddings are fused using an attention mechanism to highlight relevant objects. We then feed these conditioned features and noisy latent into a latent diffusion model to generate the object-specific audio. Finally, the latent decoder reconstructs the spectrogram, and a pre-trained HiFi-GAN vocoder generates the final audio waveform. At test time, we replace the attention with a user-provided segmentation mask, and the latent encoder for the reference spectrogram is *not* used.

contrast, our model generates sounds corresponding to user-selected objects within visual frames, without requiring explicit object segmentations and representations during training.

**Audio-visual learning.** Many works have focused on audio-visual associations due to their inherent correspondence in videos. A line of works explores the semantic correspondence, identifying which sounds and visuals are commonly associated with one another (Arandjelovic & Zisserman, 2017). This includes representation learning (Morgado et al., 2021; Huang et al., 2023a), source localization (Chen et al., 2021; Harwath et al., 2018; Chen et al., 2023), audio stylization (Chen et al., 2022a; Li et al., 2024), as well as scene classification (Chen et al., 2020; Gemmeke et al., 2017; Du et al., 2023a) and generation (Li et al., 2022b; Sung-Bin et al., 2023). Other studies leverage spatial correspondence between audio and visual streams (Owens & Efros, 2018; Korbar et al., 2018; Patrick et al., 2021) to tackle tasks like source separation (Zhao et al., 2018; 2019; Ephrat et al., 2016; Gao et al., 2018; Li et al., 2020), Foley sound synthesis (Owens et al., 2016; Du et al., 2023b), and audio spatialization (Gao & Grauman, 2019; Morgado et al., 2018; Yang et al., 2020). Inspired by these works, we aim to generate sound from the user-selected objects within visual frames.

## 3 OBJECT-AWARE SOUND GENERATION

Our goal is to generate sound from user-selected objects within a scene in a compositional way. We cast this problem by learning the correlation between audio and its corresponding visual scene and then using this correlation to predict the sound from the activated region. To achieve this, we: (i) fine-tune an off-the-shelf conditional audio generation model for sound synthesis; (ii) train an audio-guided visual object grounding model to isolate the desired object; (iii) theoretically demonstrate the equivalence between the segmentation mask and our grounding model.

### 3.1 CONDITIONAL AUDIO GENERATION MODEL

**Conditional latent diffusion model.** We adopt a pre-trained conditional latent diffusion model (Liu et al., 2023) to generate audio conditioned on textual inputs. Building upon denoising diffusion probabilistic models (Ho et al., 2020) and latent diffusion models (Rombach et al., 2022), our model operates in a compressed latent space to improve computational efficiency. Specifically, given a text prompt $t_q$ describing the desired sound and a noise vector $\epsilon \sim \mathcal{N}(\mathbf{0}, \mathbf{I})$, the model iteratively denoises the latent variables over $N$ steps to generate the corresponding audio.

Our model is trained to predict the added noise at each denoising step $n$, conditioned on the textual input $\boldsymbol{t}_q$. The training objective minimizes the difference between the predicted noise and the true noise:

$$\mathcal{L}_\theta = \mathbb{E}_{\boldsymbol{z}_0,\boldsymbol{t}_q,\boldsymbol{\epsilon}\sim\mathcal{N}(\boldsymbol{0},\mathbf{I}),n}\|\boldsymbol{\epsilon} - \boldsymbol{\epsilon}_\theta(\boldsymbol{z}_n,n,\boldsymbol{t}_q)\|_2^2 , \tag{1}$$

where $\boldsymbol{z}_0$ is the latent representation of the ground truth audio, $\boldsymbol{z}_n$ is the noisy latent at step $n$, and $\boldsymbol{\epsilon}_\theta$ is the denoising model parameterized by $\theta$.

**Mel-spectrograms compression.**    We compress mel-spectrograms into a lower-dimensional latent space using a variational autoencoder (VAE) (Kingma & Welling, 2013). The VAE encodes the mel-spectrogram $\boldsymbol{a} \in \mathbb{R}^{T \times F}$ into a latent representation $\boldsymbol{z} \in \mathbb{R}^{T' \times F' \times d}$, where $T'$ and $F'$ are reduced temporal and frequency dimensions, and $d$ is the dimensionality of the latent embeddings.

**Textual representation.**    We represent the textual input $\boldsymbol{t}_q$ using a pre-trained text encoder from CLAP (Elizalde et al., 2023), which maps the text into an embedding space $\mathcal{E}_t(\boldsymbol{t}_q) \in \mathbb{R}^L$, where $L$ denotes the embedding dimension. These text embeddings capture semantic information about the desired sound and are used to condition the diffusion model through cross-attention mechanisms (Vaswani et al., 2017).

**Classifier-free guidance.**    We employ classifier-free guidance (CFG) (Ho & Salimans, 2022) to encourage the model to learn both conditional and unconditional denoising. During training, we randomly omit the conditioning input $\boldsymbol{t}_q$ with a 10% probability. At test time, we use a guidance scale $\lambda \geq 1$ to interpolate between the conditional and unconditional predictions:

$$\tilde{\boldsymbol{\epsilon}}_\theta(\boldsymbol{z}_n,n,\boldsymbol{t}_q) = \lambda \cdot \boldsymbol{\epsilon}\theta(\boldsymbol{z}_n,n,\boldsymbol{t}_q) + (1-\lambda) \cdot \boldsymbol{\epsilon}_\theta(\boldsymbol{z}_n,n,\varnothing) , \tag{2}$$

where $\boldsymbol{\epsilon}_\theta(\boldsymbol{z}_n,n,\varnothing)$ is the unconditional prediction. This approach enhances adherence to the conditioning text while maintaining diversity in the generated audio.

**Waveform reconstruction.**    After generating the latent representation of the audio, we reconstruct the corresponding waveform. The decoder part of the VAE transforms the latent representation $\boldsymbol{z}_0$ back into a mel-spectrogram. Subsequently, a pre-trained HiFi-GAN neural vocoder (Kong et al., 2020a) is used to synthesize the time-domain audio waveform from the mel-spectrogram, producing the final audio output.

## 3.2    Text-Guided Visual Object Grounding Model

**Visual representation.**    To ground the visual objects corresponding to the desired sound, we extract features from the input image using a pre-trained visual encoder. Specifically, we utilize CLIP (Radford et al., 2021) to encode the image into a set of visual patches embeddings $\mathcal{E}_v(\boldsymbol{i}_q) \in \mathbb{R}^{P \times L}$, where $\boldsymbol{i}_q$ is the input image, $P$ is the number of patches, and $L$ denotes the embedding dimension (matching that of the text embeddings). These embeddings capture both semantic and spatial information of the visual scene.

**Scaled dot-product attention.**    We employ scaled dot-product attention (Vaswani et al., 2017) to fuse the textual and visual inputs, allowing the model to focus on specific objects within the scene. Before computing the attention, the text embeddings $\mathcal{E}_t(\boldsymbol{t}_q)$ and patch embeddings $\mathcal{E}_v(\boldsymbol{i}_q)$ are linearly projected to obtain the query, key, and value matrices. Specifically, we compute:

$$\boldsymbol{Q} = \mathcal{E}_t(\boldsymbol{t}_q)\boldsymbol{W}^Q, \quad \boldsymbol{K} = \mathcal{E}_v(\boldsymbol{i}_q)\boldsymbol{W}^K, \quad \boldsymbol{V} = \mathcal{E}_v(\boldsymbol{i}_q)\boldsymbol{W}^V, \tag{3}$$

where $\boldsymbol{W}^Q$, $\boldsymbol{W}^K$, and $\boldsymbol{W}^V$ are learnable projection matrices.

We then computes the attention weights between the projected text and each projected image patch, grounding the text in the visual domain:

$$\text{Attention}(\boldsymbol{Q},\boldsymbol{K},\boldsymbol{V}) = \text{softmax}\left(\frac{\boldsymbol{Q}\boldsymbol{K}^\top}{\sqrt{d_k}}\right)\boldsymbol{V} , \tag{4}$$

where $d_k$ is the dimensionality of the key embedding.

After obtaining the attention output, we apply an MLP layer (Murtagh, 1991) to further refine the fused representations, which enables the model to attend to image regions corresponding to the text input. In this way, we integrate the images $i_q$ with the diffusion process, allowing the model to learn to focus on the relevant regions in the image through self-supervision.

**Learnable positional encoding.** To enhance the model's ability to localize objects within the image, we incorporate learnable positional encodings (Devlin, 2018) into the attention mechanism. These encodings are added to the key and value embeddings, providing spatial information about the image patches. By learning positional information, the model can better distinguish between objects in different locations, improving grounding precision.

**Segmentation mask at test time.** After training, we have the flexibility to substitute the attention weights derived from the scaled dot-product attention with segmentation masks generated by the segment anything model (SAM) (Kirillov et al., 2023). We rescale the raw outputs of SAM into a normalized mask $m_q \in \mathbb{R}^P$, matching the mean and variance of the attention weights. This allows us to generate the desired object's sound by focusing on the regions specified by the segmentation mask. Since SAM's masks can be obtained using either text prompts or point clicks, our model supports interactive and compositional sound generation, allowing users to intuitively select objects of interest and generate their associated sounds.

### 3.3 THEORETICAL ANALYSIS

One may notice that our training pipeline uses both text and image encoders, but the test-time computation involves only the image encoder, where the softmax attention weights are replaced by the segmentation masks. This indicates an *out-of-distribution* generalization ability, where our model trained on the softmax attention weights computed by CLAP & CLIP embeddings (Equation 4) is able to generalize well on the segmentation masks computed by SAM. We hypothesize that this ability is rooted in the alignment of contrastive losses and the dot-product attention mechanism. Recall that the InfoNCE loss (Oord et al., 2018) for the text encoder in contrastive learning is given by:

$$\mathcal{L}_t\left(\mathcal{E}_t, \mathcal{E}_v\right) = \mathbb{E}_{x^T, x_{1:N}^I}\left[-\log \frac{\exp\left(\langle \mathcal{E}_v(x^T), \mathcal{E}_t(x_1^I)\rangle/\tau\right)}{\sum_{j=1}^N \exp\left(\langle \mathcal{E}_v(x^T), \mathcal{E}_t(x_j^I)\rangle/\tau\right)}\right] \tag{5}$$

where $(x^T, x_1^I)$ is the matching text-image pair, and $x_2^I, \ldots, x_N^I$ are the negative image samples associated with $x^T$. Notice that if we substitute $x^T$ with the text input $t_q$, $x_{1:N}^I$ with the image patches $i_q$, and $x_1^I$ with the matching image patch (with the text input), then the loss in Equation 5 becomes the Maximum Likelihood Estimation (MLE) loss of the softmax attention weights in Equation 4 (under proper scaling in the exponents). Therefore, the encoders $\mathcal{E}_v, \mathcal{E}_t$ are able to assign high attention weights to image patches that match with textual inputs, and low attention weights to irrelevant image patches, *working effectively as the segmentation mask at test time*. As such, the audio generation model is trained with the ability to focus only on the selected objects by segmentation masks.

In the following theorem, we formalize the above argument into a test-time error guarantee. We let $f$ denote the composition of the trained MLP layers and the audio generation model that maps an attention $a_q$ to an audio output $s_q$ on query $q$, and $v$ denote the value metric that maps a sound-image-mask tuple $(s, i, m)$ to a real number $v(s, i, m) \in \mathbb{R}$. Our goal is to bound

$$\text{err}_{\text{test}} := \mathbb{E}_q[v(f^*(p_q V^*), i_q, p_q) - v(f(m_q V), i_q, m_q)]$$

i.e., the expected (over the randomness of test query $q$) distance between the optimal value $v(f^*(p_q V^*), i_q, p_q)$ and the value of the trained model $v(f(m_q V), i_q, m_q)$ at test time. Here, $f^*$ and $V^*$ are the ground-truth counterpart of $f$ and value matrix, $p_q \in \Delta^P$ is the (normalized) ground-truth mask of query $q$ such that $p_{q,k} = \frac{\mathbb{P}(t_q|i_{q,k})}{\sum_{l=1}^P \mathbb{P}(t_q|i_{q,l})}$ for patch index $k \in \{1, \ldots, P\}$, $a_q$ represents the attention computed by Equation 4. Note that $f(m_q V)$, the audio output of the trained model, depends on the segmentation mask $m_q$ instead of the ground-truth mask $p_q$ or text input $t_q$.

**Theorem 3.1.** *Let $\epsilon_{\text{sam}} := \mathbb{E}_q[\|m_q - p_q\|_{\ell_1}]$ denote the expected $\ell_1$ error of the segmentation model. Let $\epsilon_f, \epsilon_V$ denote the expected error of $f$ and $V$ under the pre-trained CLAP & CLIP embeddings*

*respectively, and $\epsilon_{\text{contrast}}$ denote the expected contrastive loss of the encoders, more precisely,*

$$\epsilon_f = \mathbb{E}_q[v(f^*(a_q), \boldsymbol{i}_q, p_q)] - \mathbb{E}[v(f(a_q), \boldsymbol{i}_q, p_q)], \;\; \epsilon_{\boldsymbol{V}} = \|\boldsymbol{V} - \boldsymbol{V}^*\|_\infty$$

$$\epsilon_{\text{contrast}} = \mathbb{E}_{q,d\sim p_q}\left[-\log \frac{\exp\left(\langle \mathcal{E}_v(\boldsymbol{t}_q), \mathcal{E}_t(i_{q,d})\rangle_\Sigma\right)}{\sum_{k=1}^P \exp\left(\langle \mathcal{E}_v(\boldsymbol{t}_q), \mathcal{E}_t(i_{q,k})\rangle_\Sigma\right)}\right] - \mathbb{E}_{q,d\sim p_q}\left[-\log p_{q,d}\right].$$

*where $\langle\cdot,\cdot\rangle_\Sigma$ is the local inner product under $\Sigma := \boldsymbol{W}^K(\boldsymbol{W}^Q)^\top/\sqrt{d_k}$. Suppose $\|\boldsymbol{V}^*\|_\infty, \|\boldsymbol{V}\|_\infty \leq B_v$, $v$ is $L_v$-Lipschitz, and $f, f^*$ are $L_f$-Lipschitz, then we have*

$$\text{err}_{\text{test}} \leq L_v \cdot \left(L_f \cdot \left(\epsilon_{\boldsymbol{V}} + B_v \cdot \left(\epsilon_{\text{sam}} + 2\sqrt{2\epsilon_{\text{contrast}}}\right)\right) + \epsilon_{\text{sam}}\right) + \epsilon_f.$$

Due to space constraints, the proof is deferred to Appendix A.5. Theorem 3.1 implies that the test-time error can be upper bounded by the error of the pre-trained CLAP & CLIP encoders, the error of the segmentation model, and the error of the trained model under pre-trained encoders. Since the latter errors are usually small due to massive training and the regularity parameters $L_v, L_f, B_v$ are commonly modest, our method can be guaranteed to achieve high accuracy. This explains why we are able to substitute the attention weights derived from the scaled dot-product attention with segmentation masks generated by the segmentation model at test time. Our theory is further corroborated by Section 4.3, where using dot-product attention weights achieves performance on par with using segmentation masks, while additive attention fails completely.

## 4 EXPERIMENTS

### 4.1 EXPERIMENT SETUP

**Dataset.** We use the Sound-VECaps dataset (Yuan et al., 2024) as our primary data source. This dataset is derived from AudioSet (Gemmeke et al., 2017), which consists of 4,616 hours of video clips, each paired with corresponding labels and captions. To tailor the dataset for our task, we perform several preprocessing steps: (i) employ Llama (Touvron et al., 2023) to rephrase the original captions, ensuring they focus only on visible sounding objects for better consistency; (ii) exclude clips containing voiceovers and music by applying keyword-based filters such as "speech" and "music"; and (iii) train and use an off-the-shelf audio-visual matching model to retain only those videos with high correspondence scores. This reduces the dataset to 748 hours of video. Please see Appendix A.2 for more details on the dataset refinement.

**Model architecture.** Building upon the AudioLDM (Liu et al., 2023), our model integrates image inputs through a grounding model (Sec. 3.2). We employ the same VAE and HiFi-GAN vocoder, which are trained on a combination of the AudioSet (Gemmeke et al., 2017), AudioCaps (Kim et al., 2019), BBC Sound Effects (Corporation, 2017), and Freesound (Fonseca et al., 2021) datasets. The VAE is configured with a latent dimensionality $d$ of 8 channels. For embedding extraction, we utilize the "ViT-B/32" CLAP audio encoder (Elizalde et al., 2023) and the CLIP image encoder (Radford et al., 2021). These embeddings are then incorporated into the U-Net-based diffusion model through cross-attention (Vaswani et al., 2017). We implement a linear noise schedule consisting of $N = 1000$ diffusion steps, from $\beta_1 = 0.0015$ to $\beta_N = 0.0195$. The DDIM sampling method (Song et al., 2020) is used with 200 steps to facilitate efficient generation. At test time, we apply CFG with a guidance scale $\lambda$ set to 2, as defined in Equation 2.

**Training configuration.** To facilitate parallel training, each video's soundtrack is either truncated or zero-padded to achieve a fixed duration of 10 seconds and then converted to a 16 kHz sample rate in 32-bit floating-point PCM format. We apply a 512-point discrete Fourier transform with a frame length of 64 ms and a frame shift of 10 ms. For each video, a single visual frame is randomly chosen to serve as the input image. The model is then trained using the AdamW optimizer (Loshchilov & Hutter, 2017) with a batch size of 64, a learning rate of $10^{-4}$, $\beta_1 = 0.95$, $\beta_2 = 0.999$, $\epsilon = 10^{-6}$, and a weight decay of $10^{-3}$ over 300 epochs.

**Evaluation metrics.** We use both quantitative and qualitative metrics (see Appendix A.3 for more evaluation details) to evaluate the performance of our model. For the objective evaluation, we employ several metrics, including Sound Event Accuracy (ACC), which leverages the PANNs model (Kong et al., 2020b) to predict and sample sound event logits based on the annotated labels and then

| Method | ACC (↑) | FAD (↓) | KL (↓) | IS (↑) | AVC (↑) | OVL (↑) | RET (↑) | REI (↑) | REO (↑) |
|---|---|---|---|---|---|---|---|---|---|
| Ground Truth | / | / | / | / | 0.962 | 4.12 ± 0.06 | 4.02 ± 0.05 | 4.06 ± 0.07 | / |
| AudioLDM 1 | 0.314 | 3.761 | 1.542 | 1.541 | 0.701 | 2.76 ± 0.03 | 3.08 ± 0.07 | 2.88 ± 0.02 | 2.12 ± 0.03 |
| AudioLDM 2 | 0.502 | 2.981 | 1.141 | 1.785 | 0.747 | 2.97 ± 0.02 | 3.21 ± 0.04 | 3.06 ± 0.04 | 2.44 ± 0.02 |
| Make-an-Audio | 0.309 | 3.555 | 1.443 | 1.673 | 0.712 | 2.74 ± 0.08 | 3.06 ± 0.05 | 2.89 ± 0.05 | 2.08 ± 0.04 |
| Im2Wav | 0.499 | 3.602 | 1.526 | 1.872 | 0.798 | 2.88 ± 0.05 | 3.12 ± 0.04 | 3.01 ± 0.05 | 2.48 ± 0.06 |
| SpecVQGAN | 0.611 | 2.515 | 1.142 | 1.965 | 0.825 | 2.94 ± 0.04 | 3.26 ± 0.03 | 3.11 ± 0.06 | 2.51 ± 0.04 |
| Diff-Foley | 0.683 | 1.908 | 0.783 | 2.010 | 0.842 | 3.09 ± 0.06 | 3.43 ± 0.05 | 3.32 ± 0.03 | 2.52 ± 0.06 |
| Ours | **0.859** | **1.271** | **0.517** | **2.102** | **0.891** | **3.31 ± 0.04** | **3.62 ± 0.05** | **3.48 ± 0.04** | **3.74 ± 0.07** |

Table 1: Quantitative comparison of our method and baselines across different metrics. The subjective OVL, RET, REI, and REO scores are presented with 95% confidence intervals.

compute the mean accuracy across the dataset. We also measure the semantic alignment between the output and target using three established metrics: (i) Fréchet Audio Distance (FAD) (Kilgour et al., 2019), which quantifies how close the generated audio is to the real audio in latent space; (ii) Kullback-Leibler Divergence (KL), which assesses the alignment of distributions between the generated and target audio; and (iii) the Inception Score (IS) (Salimans et al., 2016), which evaluates the diversity of the generated audio. Additionally, the *Audio-Visual Correspondence* (AVC) (Arandjelovic & Zisserman, 2017) is used to measure the semantic coherence between the input image and the resulting audio, indicating how well the sounds match the visual context. We report this using the average cosine similarity of features extracted by OpenL3 (Cramer et al., 2019).

For subjective evaluation, we conduct a human study to assess the quality and relevance of the generated audio. We present both the holistic samples and the object-selected samples. Each participant is provided with an input image, along with the corresponding generated audio, and is asked to rate each sample on a scale from 1 to 5 based on several criteria: (i) Overall Quality (OVL), which evaluates the general quality of the audio; (ii) Relevance to the Text Prompt (RET), which assesses how well the audio matches any associated text description; (iii) Relevance to the Input Image (REI), which judges the alignment between the audio and the visual content; and Relevance to the Selected Object (REO), which focuses on how well the generated audio aligns with a specific object in the visual scene.

**Baselines.** We compare our method with several baseline models, each of which is adapted for our task:

- **AudioLDM 1 & 2** (Liu et al., 2023; 2024): These models are originally designed for text-to-audio generation, but we modify them by swapping their text embeddings with image embeddings. We fine-tune these models on our dataset for a fair comparison.
- **Make-an-Audio** (Huang et al., 2023b): Make-an-Audio supports either text or image prompts for sound generation. We extract its image-based branch and fine-tune it on our dataset.
- **Im2Wav** (Sheffer & Adi, 2023): Im2Wav is an image-guided open-domain audio generation model that operates auto-regressively. Since the original model generates only 4 seconds of audio, we retrain it on our dataset to adapt it to our task.
- **SpecVQGAN** (Iashin & Rahtu, 2021): SpecVQGAN is a two-stream VQGAN model (Esser et al., 2021) designed for video-to-audio generation. We modify it by randomly sampling a single frame from video data and fine-tune it for our task.
- **Diff-Foley** (Luo et al., 2023): Diff-Foley is a diffusion model that generates sound semantically and temporally aligned with the video. Similar to SpecVQGAN, we fine-tune it on our dataset using randomly sampled video frames.

### 4.2 COMPARISON TO BASELINES

**Quantitative results.** Table 1 compares our approach against the baselines on the Sound-VECaps dataset. Our model outperforms the baselines across different metrics, highlighting its ability to produce high-quality audio. In particular, our method achieves the best ACC metrics, indicating its capacity to generate sound closely linked to the visual objects in the scene. Diff-Foley shows competitive performance among the baselines, likely due to its contrastive representations, which map visual and audio features to a shared latent space, improving audio-visual consistency. Although Im2Wav and SpecVQGAN achieve reasonable AVC scores, they struggle with FAD and KL, indi-

Figure 3: **Qualitative model comparison**. We show sound generation results for our method and the baselines, each of which is conditioned on an image, text, or segmentation mask.

cating they fall short in generating high-quality sounds. Similarly, AudioLDM and Make-an-Audio show relatively lower accuracy and semantic alignment, which could be due to their original design for the text-to-audio task rather than the image-guided one. Notably, our model significantly surpasses Diff-Foley in terms of FAD and KL, suggesting that it can generate audio that is not only realistic but also semantically linked to the visual inputs. These results indicate the advantage of our method in leveraging visual cues for more contextually relevant sound generation.

For subjective evaluation, we randomly select 100 generated samples from the test set, with 50 of them manually processed to create segmentation masks for specific objects within a scene. These samples are then rated by 50 participants. Our model receives the highest average ratings across all subjective measures, with a particularly notable lead in REO, suggesting that it generates sounds aligned with the objects in the image. Interestingly, we observe that all the baselines achieve relatively close scores for REO, which demonstrates that our method is particularly good at linking audio to object-level visual cues, a feature that is less evident in the baselines. Moreover, participants consistently rated the OVL, RET, and REI of our model higher, further validating the objective metrics and highlighting its improved contextual alignment.

**Qualitative results.** Figure 3 compares our method with the baselines on the Sound-VECaps dataset. In the first example, where both a dog and a goose are present, all baselines only generate dog growls, missing the goose honks. Our method, however, captures both sounds, illustrating its object-aware capability. Similarly, in the second and third examples, involving a car with distant chatter and a train with people talking, the baselines produce either one of the sound events but not all simultaneously. By contrast, our model successfully generates the complete soundscape. The final example presents a small jet in the background with the crowd cheering. Vision-based models fail to detect the jet due to the jet's small size in the image, generating only the crowd and wind noises, while text-based models struggle to combine multiple sounds. Our approach accurately captures all relevant sounds, highlighting its ability to generate accurate sounds aligned with complex visual scenes. For a more direct experience, please view the results video in the supplement and on the project webpage.

### 4.3 Ablation Study and Analysis

Table 2 summarizes the ablation experiments. We explore the following model variations: (i) freezing the latent diffusion weights rather than fine-tuning them; (ii) replacing single-head attention with multi-head attention; (iii) substituting text-image attention with audio-image attention; (iv) altering the attention mechanism from dot-product to additive attention; and (v) using text-image attention instead of segmentation masks during inference. We also show additional results in Appendix A.4.

**Effect of freezing diffusion weights.** We test the impact of freezing the latent diffusion model weights instead of fine-tuning them during training. We observe that freezing the weights degrades the performance, which suggests that fine-tuning is required to achieve more coherent audio.

**Impact of attention head.** We compare our single-head attention mechanism with the multi-head counterpart (Vaswani et al., 2017). The multi-head approach enhances the alignment between tex-

| Method | ACC (↑) | FAD (↓) | KL (↓) | IS (↑) | AVC (↑) |
|---|---|---|---|---|---|
| (i) Frozen Diffusion | 0.692 | 1.543 | 1.047 | 1.943 | 0.733 |
| (ii) Multi-Head Attention | 0.415 | 2.238 | 1.903 | **2.115** | 0.887 |
| (iii) Audio-Image Attention | 0.634 | 1.761 | 1.232 | 1.731 | 0.692 |
| (iv) Additive Attention | 0.103 | 15.747 | 7.425 | 1.343 | 0.137 |
| (v) Text-Image Attention | 0.856 | **1.270** | 0.520 | 2.097 | 0.890 |
| Ours | **0.859** | 1.271 | **0.517** | 2.102 | **0.891** |

Table 2: Quantitative ablation studies on the Sound-VECaps dataset.

tual inputs and the generated audio, leading to a stronger correspondence between text descriptions and sound outputs. However, this improvement reduces controllability when specifying specific audio characteristics based on the segmentation mask. We conjecture that this limitation arises because each head in the multi-head attention focuses on different regions of the input (Voita et al., 2019; Hamilton et al., 2024). While this strategy increases text-audio alignment, the lack of a clear definition for each head's specific scope reduces the interpretability of the final results. This likely contributes to the masking results deviating from expectations.

**Choice of attention modality.** We assess the effectiveness of text-image attention compared to audio-image attention. The audio-image attention variant shows a decline in performance, which could be attributed to the inherent limitations of the CLAP model in representing overlapping audios. This limitation probably introduces noise, thereby weakening the model's ability to form audio-visual associations essential for sound generation.

**Evaluation of attention scoring mechanism.** We investigate the role of the attention scoring function by replacing dot-product attention with the additive one (Bahdanau, 2014). The additive attention variant collapses significantly, indicating that segmentation masks are not a suitable replacement for this attention. Explained by the theory in Section 3.3, this could be because addition operations are not compatible with the contrastive losses used by CLAP & CLIP and segmentation masks generated by SAM, which disrupts our grounding model.

**Role of segmentation masks during inference.** We compare the standard text-image attention mechanism to the proposed segmentation masks at test time. The results show that text-image attention achieves performance on par with the segmentation mask approach (ours). This suggests that both methods provide similar levels of spatial and semantic guidance for audio generation. This finding also supports the theory discussed in Section 3.3.

### 4.4 CROSS-DATASET EVALUATION

**Visualization between grounding and masking.** In Figure 4, we visualize the comparison between the attention maps generated by our model and the segmentation masks produced by SAM. For this, we use images from Places (Zhou et al., 2017) and text prompts derived from BLIP (Li et al., 2022a). To visualize the attention maps, we apply bilinear interpolation to match the resolution of the segmentation masks. Our results show a strong alignment between our model's attention maps and the segmentation masks, providing empirical support for the theoretical analysis in Section 3.3 and the findings of the ablation study in Section 4.3. While the segmentation masks represent a form of "hard" attention, directly highlighting specific regions, our model produces "soft" attention maps that provide a probabilistic focus on the relevant areas. This similarity indicates that, through training, our model effectively learns to capture object-specific regions similar to those identified by segmentation, achieving the desired grounding in a flexible manner. Furthermore, this

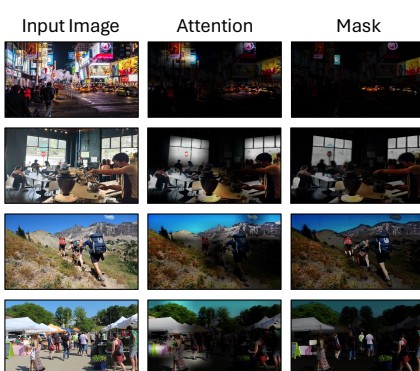

Figure 4: **Visualization results**. We visualize the difference between attention maps and segmentation masks using images from Places (Zhou et al., 2017) and text prompts from BLIP (Li et al., 2022a).

observation suggests that attention maps can be replaced with segmentation masks at test time.

**Compositional sound generation.** We ask whether our model will generate object-specific sounds by isolating individual objects within a scene. As shown in Figure 5, we use the same image for each scene, separating different objects (cars, people, seagulls, etc.) to generate corresponding audio outputs. The results illustrate that our model successfully

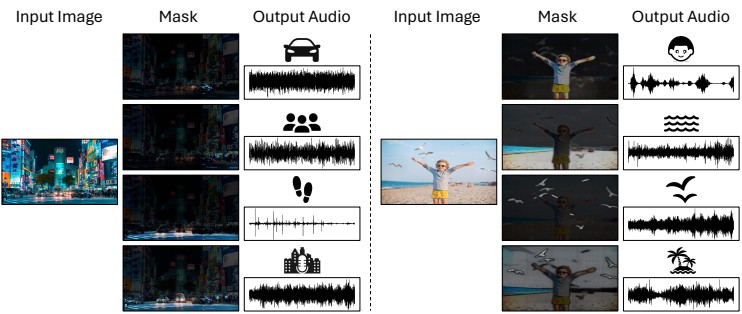

Figure 5: **Compositional sound generation**. Our model generates object-specific sounds in the city (left) and beach (right) scenes, and composes a complete soundscape when multiple objects are selected.

learns to generate distinct sounds for each object, such as car engines or footsteps, reflecting their unique sound textures. Furthermore, when multiple objects are selected together, the model compositionally generates the entire soundscape that represents the scene property. This capability highlights our model's strength in decomposing and synthesizing audio-visual elements for sound generation.

**Sound adaptation to visual texture changes.** We explore whether our method can generate soundscapes that adapt to changes in visual textures, inspired by audio-visual video editing (Lee et al., 2023). Starting with images from the Places (Zhou

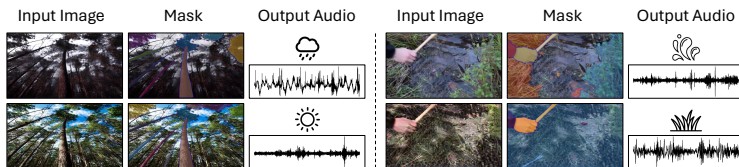

Figure 6: **Generating soundscapes from visual texture changes.**. We generate different soundscapes by manipulating the visual textures of the same scene, such as changing weather (left) or materials (right).

et al., 2017) and Greatest Hits (Owens et al., 2016) datasets, we apply an off-the-shelf image translation model (Park et al., 2020; Li et al., 2022b) to create paired scenes (e.g., sunny-rainy, water-grass), and then overlay full-image segmentation masks on top. As illustrated in Figure 6, our model generates context-appropriate soundscapes. For instance, it generates rain sounds for dark skies, wind sounds for clear skies, water splashing for watery surfaces, and grass crunching for grassy areas. This demonstrates that our model successfully captures variations in visual textures to generate corresponding audio.

## 5 CONCLUSION

In this paper, we proposed an *object-aware sound generation* model, focusing on aligning generated sounds with specific visual objects in complex scenes. To achieve this, we developed a diffusion model grounded in object-centric representations, enhancing the association between objects and their corresponding sounds. Our theoretical analysis demonstrates that the object-grounding mechanism is functionally equivalent to segmentation masks. Quantitative and qualitative evaluations show that our model surpasses baselines in sound-object alignment, enabling cross-dataset generalization and compositional sound generation. We hope this work not only advances controllable sound generation but also inspires further exploration into the relationships between objects and soundscapes.

**Limitations and broader impacts.** Our model shows promising results in generating object-specific sounds from images but has certain limitations. First, since our model relies on static images, it may struggle to produce non-stationary audio synchronizing with dynamic events, such as impact sounds (Figure 6). Additionally, it may lack precise control over the type of sound produced for an object, leading to potential ambiguity. For example, a car might be associated with various sounds, such as siren or engine noise (Figure 3). Lastly, while useful for content creation like filmmaking, our model also poses a potential risk, as it could be exploited to create misleading videos.

ETHICS STATEMENT

This paper introduces an *object-aware sound generation* model. It is trained on publicly available datasets, such as AudioSet and Sound-VECaps, which do not contain personally identifiable information. We have taken steps to ensure compliance with data usage policies, and our model does not involve human subjects or raise privacy concerns. We believe our work poses minimal ethical risks, as it focuses on enhancing sound-object alignment in a controlled research environment. However, we encourage responsible use of our model, particularly when applied to real-world scenarios.

REPRODUCIBILITY STATEMENT

To facilitate the reproducibility of our results, we provide detailed information in multiple sections of this paper and its appendix. A comprehensive description of the dataset is presented in Section 4.1 of the main paper, with additional data refinement details included in Appendix A.2. The key training configurations, including hyperparameters, are outlined in Section 4.1. Our proposed method is illustrated in Section 3, and the source code has been made available in the supplement for reference.

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

## A.1 RESULTS VIDEO

Our results video provided in the supplement, as well as on the project webpage, showcases our model's ability to generate accurate sound textures based on the mask prompts. Specifically, the video demonstrates:

- Our model can compositionally generate object-specific sounds within complex scenes.
- Despite being trained on the Sound-VECaps dataset (Yuan et al., 2024), our model can be successfully applied to out-of-domain visual scenes, including those from the Places dataset (Zhou et al., 2017), the Greatest Hits dataset (Owens et al., 2016), and even random web images.
- Our model can capture variations in visual textures to generate corresponding audio.

## A.2 DATASET REFINEMENT

We use the Sound-VECaps dataset (Yuan et al., 2024), derived from AudioSet (Gemmeke et al., 2017), as the primary source for this task. The original dataset comprises 4,616 hours of video clips, each paired with corresponding labels and captions. To adapt this dataset for our use, we apply the following refinement steps.

**Audio-visual matching.** To ensure strong correspondence between audio and visual inputs, we train an audio-visual matching model (Figure 8), which consists of a 6-layer non-causal transformer with a rotary positional embedding mechanism (Su et al., 2024). Visual embeddings are extracted using the ViT-B/16 Transformer module from CLIP (Radford et al., 2021), while audio embeddings are generated using the BEATs model (Chen et al., 2022b). Both embeddings are then passed through a 3-layer MLP to match a 768-dimensional space. The model is trained in a self-supervised manner (Owens & Efros, 2018; Korbar et al., 2018), treating audio-visual pairs from the same temporal instance as matches and those from different videos as mismatches, which allows the model to learn audio-visual correspondences without human annotations.

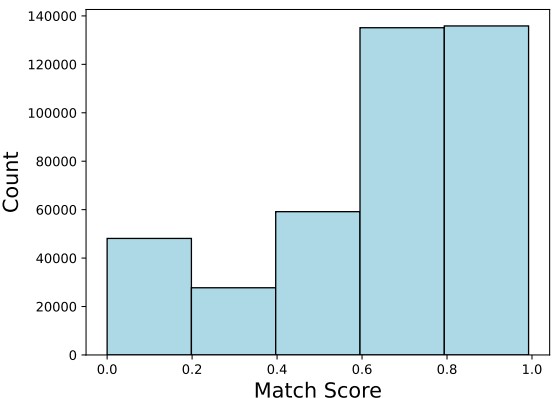

Figure 7: **Distribution of matching scores.** We present the scores for audio-visual pairs in the Sound-VECaps dataset.

For training efficiency, the videos are standardized to 8 frames per second, with each frame resized to 224x224 pixels. During the evaluation, our model achieves an accuracy of 91% for matching scenarios and 85% for non-matching scenarios on a set of 100 matched and 100 mismatched samples, indicating its effectiveness in capturing audio-visual alignment. We use this model to score each clip in the Sound-VECaps dataset, with results shown in Figure 7. A threshold of 0.6 is then applied to filter the dataset.

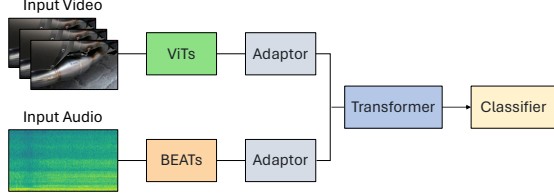

Figure 8: **Architecture of the audio-visual matching model.** We train a model to quantify the correspondence between a video and its corresponding soundtrack.

**Caption rephrasing.** To ensure captions focus exclusively on visible sounding objects, we utilize Llama (Touvron et al., 2023) with a tailored prompt (Figure 9). Given the video and audio captions, our prompt instructs the model to generate a single sentence highlighting the common features between the audio and visual content. The prompt emphasizes including only events present in both modalities, while excluding modality-specific details such as overly specific visual features. The

**Role-System:**
You are a helpful assistant for identifying audio-visual events and generating sentences. Your task is to identify the overlapping or common features between a 10-second audio and the corresponding visual description, and help the user to generate a single sentence of caption that represents this intersection.
The caption feature is a sentence generated by an audio-caption model: **{enclap_caption}**.
The label feature is several audio events that happened in the audio: **{audio_label}**.
Lastly, the user is given several sentences which are the image description of the scene for each second, connected by "and then".
Please identify all the audio events and visual elements based on all three features and try to conclude in one single sentence to describe this scene with the shared audio-visual events or actions that present sound and sight together.
Please emphasize time features to present the order of each event, such as "and then", "followed by", "after" for order; "and", "while" etc., for parallel events.
**Intersection Focus:**
• Based on the first caption feature, you might need to change or alter any wrong audio event, improve the sentence with more features, such as the weather, the emotion of any people, the description of the car and so on.
• Keep only the features that are common between the audio and visual descriptions. If an event or element is mentioned in both the audio and the visual description, include it in the final caption.
• Omit any feature or detail that is present in only one modality. This includes removing overly specific visual details, such as the color, shape, any text or label, name and what people are writing and so on, that do not align with the audio description and vice versa.
Please ensure that the final caption accurately reflects the common elements of the audio-visual scene, maintaining the order of occurrence, and capturing the shared background, foreground, and context.
**Role-User:**
The descriptions of the frames are: **{frame_caption}**

Figure 9: **Prompt for Llama**. We extract common features between the audio and visual caption using Llama, ensuring the resulting caption focuses on events present in both modalities while avoiding overly specific details.

model is guided to capture the order and parallel occurrence of events using temporal markers like "and then," "followed by," and "while." This process enhances the consistency between audio and visual descriptions.

**Audio filtering.** We filter out clips containing human vocalizations (e.g., singing, talking), voiceovers, and music using a sound event detection model (Kong et al., 2020b) and the metadata from AudioSet. This step ensures that the remaining audio data largely consists of ambient and context-specific sounds that are more likely to align with the visual content.

After applying these refinement steps, the resulting data is reduced to 748 hours of video clips that exhibit high audio-visual correspondence.

## A.3 ADDITIONAL EVALUATION DETAILS

**ACC.** We use the PANNs model (Kong et al., 2020b) to compute ACC for each audio clip, leveraging annotations provided by AudioSet. First, we process each audio clip through the pre-trained PANNs model to obtain the logit values for all possible sound event classes. Using the AudioSet annotations, we then sample the logits corresponding to the annotated labels for each clip. Since these logits are the softmax outputs, they represent the model's confidence for each event, allowing us to interpret them as accuracy scores for the labeled events. We then compute the mean of these sampled logits across all clips in the dataset to obtain the final ACC score.

**FAD, KL, and IS.** We measure FAD, KL, and IS using the AudioLDM-Eval toolbox[1]. The reference and generated audio files are organized into separate folders, and the toolbox is run in paired mode.

**AVC.** We measure AVC using a two-stream network Arandjelovic & Zisserman (2017). One stream extracts audio features, while the other extracts visual features. We use OpenL3 Cramer et al. (2019) to obtain these features and compute the cosine similarity for each image-audio pair. Specifically, we employ the "env" content type model with a 512-dimensional linear spectrogram representation.

**Human evaluation.** We conducted a human evaluation to assess the quality and relevance of the generated audio using Amazon Mechanical Turk. The interface for this study is shown in Figure 10. Each participant was presented with an input image and the corresponding generated audio, then

---
[1]https://github.com/haoheliu/audioldm_eval

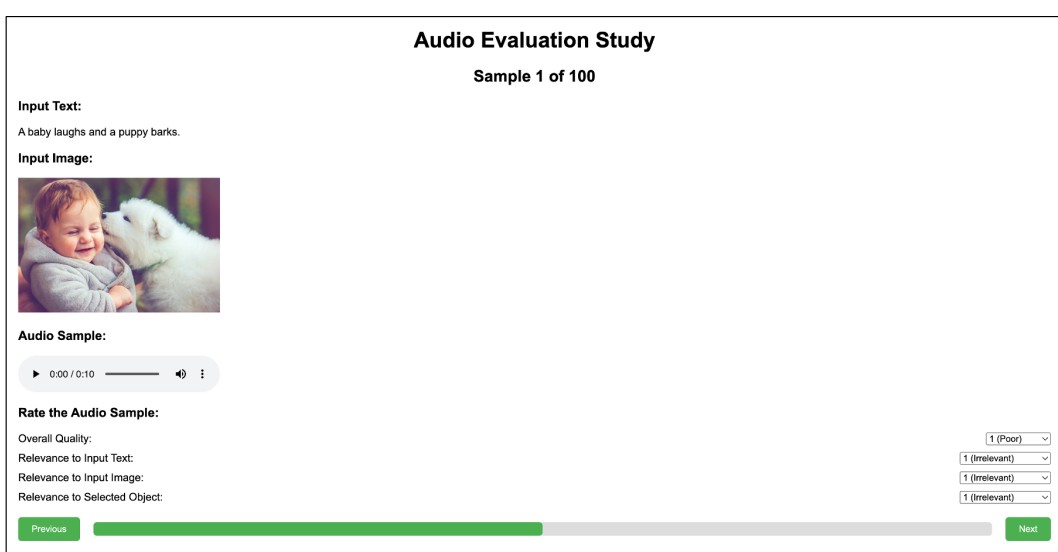

Figure 10: **Human evaluation interface.** We show the interface used for the subjective evaluation of generated audio samples. Participants are presented with input text, an image, and a corresponding audio sample, and are instructed to rate the audio on four criteria. All ratings must be completed before advancing to the next sample.

| Scale | ACC (↑) | FAD (↓) | KL (↓) | IS (↑) | AVC (↑) |
|---|---|---|---|---|---|
| $\lambda = 1.0$ | 0.413 | 2.021 | 0.914 | 1.336 | 0.674 |
| $\lambda = 1.5$ | 0.657 | 1.558 | 0.762 | 1.617 | 0.751 |
| $\lambda = 2.0$ | **0.859** | **1.271** | **0.517** | **2.102** | **0.891** |
| $\lambda = 2.5$ | 0.807 | 1.440 | 0.589 | 2.012 | 0.853 |
| $\lambda = 3.0$ | 0.796 | 1.482 | 0.576 | 2.023 | 0.841 |

Table 3: Quantitative results under different CFG scales.

rated each sample on a scale from 1 to 5 based on the following criteria: (i) Overall Quality (OVL), assessing the general audio quality; (ii) Relevance to Input Text (RET), measuring the alignment of the audio with the associated text description; (iii) Relevance to Input Image (REI), evaluating how well the audio corresponds to the visual content; and (iv) Relevance to Selected Object (REO), focusing on the alignment of the audio with a specific object in the image.

We randomly selected 100 samples for evaluation, each rated by 50 unique participants to ensure reliability. The samples included both holistic and object-specific audio. To control for random responses, we incorporated a set of noise-only samples. Consistently low scores for these control samples confirmed the reliability of participants. Additionally, we ensured that each participant spent at least 90 seconds evaluating each sample to guarantee thoughtful assessment.

To further validate our results, we computed the inter-rater reliability using Cohen's kappa (McHugh, 2012), which indicated a substantial agreement among raters ($\kappa = 0.78$). Furthermore, we conducted a statistical significance test (paired t-test) (Kim, 2015) between our model and baselines for each criterion, confirming that the improvements reported are statistically significant ($p < 0.01$). The final scores presented in the main paper are the mean ratings across all participants.

## A.4 ADDITIONAL RESULTS

**Different CFG scales.** We evaluate our model's performance across CFG scales ranging from 1.0 to 3.0. As shown in Table 3, there is a consistent improvement in metrics as $\lambda$ increases from 1.0 to 2.0, reaching peak performance at $\lambda = 2.0$. However, further increasing $\lambda$ beyond 2.0 results in a gradual decline across most metrics.

**Different thresholds of audio-visual matching.** We test our model's performance across different audio-visual matching thresholds, varying from 0.4 to 0.8 (Figure 7). The same held-out test set is

| Threshold | ACC ($\uparrow$) | FAD ($\downarrow$) | KL ($\downarrow$) | IS ($\uparrow$) | AVC ($\uparrow$) |
|---|---|---|---|---|---|
| 0.4 | 0.521 | 1.874 | 0.888 | 1.432 | 0.696 |
| 0.5 | 0.743 | 1.536 | 0.691 | 1.625 | 0.774 |
| 0.6 | **0.859** | **1.271** | **0.517** | **2.102** | **0.891** |
| 0.7 | 0.845 | 1.387 | 0.612 | 1.987 | 0.882 |
| 0.8 | 0.812 | 1.501 | 0.664 | 2.005 | 0.879 |

Table 4: Quantitative results under different audio-visual matching scores.

| Method | ACC ($\uparrow$) | FAD ($\downarrow$) | KL ($\downarrow$) | IS ($\uparrow$) | AVC ($\uparrow$) |
|---|---|---|---|---|---|
| w/o PE | 0.787 | 1.493 | 0.674 | 1.913 | 0.779 |
| w/ PE | **0.859** | **1.271** | **0.517** | **2.102** | **0.891** |

Table 5: Comparison of model performance with and without positional encoding.

used to assess the metrics, with results presented in Table 4. We empirically find that the model achieves optimal performance at a threshold of 0.6.

**Effect of positional encoding.** We assess the impact of positional encoding (PE) on our model's performance. As shown in Table 5, removing positional encoding leads to a significant degradation across all metrics, highlighting its importance in the model's overall performance.

## A.5 PROOF OF THEOREM 3.1

*Proof.* For notation simplicity, let $u_q \in \Delta^P$ denote the softmax attention weight computed on query $q$ such that $u_{q,l} = \frac{\exp(\langle \mathcal{E}_v(\boldsymbol{t}_q), \mathcal{E}_t(\boldsymbol{i}_{q,l}) \rangle_\Sigma)}{\sum_{k=1}^{P} \exp(\langle \mathcal{E}_v(\boldsymbol{t}_q), \mathcal{E}_t(\boldsymbol{i}_{q,k}) \rangle_\Sigma)}$. We first state the following lemma.

**Lemma A.5.1.** *Under the same conditions in Theorem 3.1, we have*

$$\mathbb{E}_q[\|u_q - p_q\|_{\ell_1}] \leq \sqrt{2\epsilon_{\text{contrast}}}$$

*Proof.* For notation simplicity, let $u_q \in \Delta^P$ denote the attention mask computed on query $q$ such that $u_{q,l} = \frac{\exp(\langle \mathcal{E}_v(\boldsymbol{t}_q), \mathcal{E}_t(\boldsymbol{i}_{q,l}) \rangle_\Sigma)}{\sum_{k=1}^{P} \exp(\langle \mathcal{E}_v(\boldsymbol{t}_q), \mathcal{E}_t(\boldsymbol{i}_{q,k}) \rangle_\Sigma)}$. Notice that

$$\epsilon_{\text{contrast}} = \mathbb{E}_{q,d \sim p_q} \left[ -\log \frac{\exp\left( \langle \mathcal{E}_v(\boldsymbol{t}_q), \mathcal{E}_t(\boldsymbol{i}_{q,d}) \rangle_\Sigma \right)}{\sum_{k=1}^{P} \exp\left( \langle \mathcal{E}_v(\boldsymbol{t}_q), \mathcal{E}_t(\boldsymbol{i}_{q,k}) \rangle_\Sigma \right)} \right] - \mathbb{E}_{q,d \sim p_q} \left[ -\log p_{q,d} \right]$$

$$= \mathbb{E}_{q,d \sim p_q} \left[ \log \frac{p_{q,d}}{u_{q,d}} \right]$$

$$= \mathbb{E}_q \left[ D_{\text{KL}}(p_{q,d}, u_{q,d}) \right]$$

where $D_{\text{KL}}$ denotes the KL distance. By Pinsker's inequality and Cauchy-Schwarz inequality,

$$\epsilon_{\text{contrast}} = \mathbb{E}_q \left[ D_{\text{KL}}(p_{q,d}, u_{q,d}) \right]$$

$$\geq \frac{1}{2} \cdot \mathbb{E}_q \left[ \|p_{q,d} - u_{q,d}\|_{\ell_1}^2 \right]$$

$$\geq \frac{1}{2} \cdot \left( \mathbb{E}_q \left[ \|p_{q,d} - u_{q,d}\|_{\ell_1} \right] \right)^2 .$$

It follows that

$$\mathbb{E}_q[\|u_q - p_q\|_{\ell_1}] \leq \sqrt{2\epsilon_{\text{contrast}}}.$$

$\square$

Returning to the proof of Theorem 3.1, let $s_q := f(a_q) = f(u_q \boldsymbol{V})$ denote the audio output on query $q$ by the trained model. We decompose $\mathrm{err}_{\mathrm{test}}$ by

$$
\begin{aligned}
&\mathrm{err}_{\mathrm{test}} \\
&= \underbrace{\mathbb{E}_q[v(f^*(p_q\boldsymbol{V}^*), \boldsymbol{i}_q, p_q)] - \mathbb{E}_q[v(f^*(u_q\boldsymbol{V}^*), \boldsymbol{i}_q, p_q)]}_{A} + \underbrace{\mathbb{E}_q[v(f^*(u_q\boldsymbol{V}^*), \boldsymbol{i}_q, p_q)] - \mathbb{E}_q[v(f^*(a_q), \boldsymbol{i}_q, p_q)]}_{B} \\
&\quad + \underbrace{\mathbb{E}_q[v(f^*(a_q), \boldsymbol{i}_q, p_q)] - \mathbb{E}_q[v(f(a_q), \boldsymbol{i}_q, p_q)]}_{C} + \underbrace{\mathbb{E}_q[v(f(a_q), \boldsymbol{i}_q, p_q)] - \mathbb{E}_q[v(f(a_q), \boldsymbol{i}_q, \boldsymbol{m}_q)]}_{D} \\
&\quad + \underbrace{\mathbb{E}_q[v(f(a_q), \boldsymbol{i}_q, \boldsymbol{m}_q)] - \mathbb{E}_q[v(f(\boldsymbol{m}_q\boldsymbol{V}), \boldsymbol{i}_q, \boldsymbol{m}_q)]}_{E}.
\end{aligned}
$$

By Lemma A.5.1 and $\|\boldsymbol{V}^*\|_\infty \leq B_v$, we have

$$
\begin{aligned}
A &\leq \mathbb{E}_q[L_v \cdot L_f \cdot B_v \cdot \|u_q - p_q\|_{\ell_1}] \\
&\leq L_v \cdot L_f \cdot B_v \cdot \sqrt{2\epsilon_{\mathrm{contrast}}}.
\end{aligned}
$$

Since $\|\boldsymbol{V}^* - \boldsymbol{V}\|_\infty \leq \epsilon_v$ and $\|u_q\|_1 = 1$, we have

$$
\begin{aligned}
B &= \mathbb{E}_q[v(f^*(u_q\boldsymbol{V}^*), \boldsymbol{i}_q, p_q)] - \mathbb{E}_q[v(f^*(u_q\boldsymbol{V}), \boldsymbol{i}_q, p_q)] \\
&\leq L_v \cdot L_f \cdot \epsilon_{\boldsymbol{V}}.
\end{aligned}
$$

By definition, $C \leq \epsilon_f$. Using the definition $\epsilon_{\mathrm{sam}} = \mathbb{E}_q[\|\boldsymbol{m}_q - p_q\|_{\ell_1}]$, we have

$$
\begin{aligned}
D &\leq \mathbb{E}_q[L_v \cdot \|\boldsymbol{m}_q - p_q\|_{\ell_1}] \\
&\leq L_v \cdot \epsilon_{\mathrm{sam}}.
\end{aligned}
$$

and using $\|\boldsymbol{V}\|_\infty \leq B_v$ with Lemma A.5.1,

$$
\begin{aligned}
E &\leq \mathbb{E}_q[L_v \cdot L_f \cdot B_v \cdot \|\boldsymbol{m}_q - u_q\|_{\ell_1}] \\
&\leq L_v \cdot L_f \cdot B_v \cdot (\epsilon_{\mathrm{sam}} + \sqrt{2\epsilon_{\mathrm{contrast}}}).
\end{aligned}
$$

Combining, we have

$$
\mathrm{err}_{\mathrm{test}} \leq L_v \cdot \left( L_f \cdot \left( \epsilon_{\boldsymbol{V}} + B_v \cdot \left( \epsilon_{\mathrm{sam}} + 2\sqrt{2\epsilon_{\mathrm{contrast}}} \right) \right) + \epsilon_{\mathrm{sam}} \right) + \epsilon_f.
$$

This completes the proof. $\qquad\square$

