# OpenReview forum: "Object-Aware Audio-Visual Sound Generation"
_ICLR.cc/2025/Conference — ICLR 2025 Conference Withdrawn Submission_

### Official Review · Reviewer_TZCS · 2024-10-31

**Soundness:** 2
**Presentation:** 3
**Contribution:** 2
**Rating:** 3
**Confidence:** 4

**Summary:**

This paper proposes a framework to generate sound from visual input by selectively attending to masked regions in images. Using a pre-trained conditional latent diffusion model as a backbone, the proposed framework learns to integrate a soft attention mask with visual features, which is replaced with a hard segmentation mask from SAM during inference. The effectiveness of the proposed framework is evaluated on a newly curated dataset based on Sound-VECaps, where it outperforms a wide range of prior art, both qualitatively and quantitatively.

**Strengths:**

- The proposed framework proposes a plausible way to integrate a strong off-the-shelf audio generation model, AudioLDM, for image-to-audio generation.
- Evaluation metrics cover diverse aspects of ensuring the quality of audio generation, and the qualitative evaluation is conducted rigorously.
- Generated audio samples on the project page are visually relevant and sound natural.

**Weaknesses:**

- I'm not convinced that the proposed framework can legitimately claim "compositional generation" of audio from visual input, a term frequently used in the draft. The framework generates output audio directly from hard/soft masks, rather than considering combinations or selections of different components in the visual input. For instance, the generated sound for a specific object may vary significantly between using a full scene versus partial objects as input, and the model appears to lack control over this variation.
- The theoretical analysis in Section 3.3 appears to have been retrofitted to match the heuristics addressing the gap between training and inference, lacking sufficient rigor. The validity of the value metric in Line 258 is not adequately justified, and the claim that bounding the expectation of value differences leads to test-time generalization remains unclear. Furthermore, the Lipschitz assumption represents an especially strict condition for deep/foundational neural networks, and its adoption without thorough justification is questionable. Most critically, the statement in Line 283 that "Since errors are _usually_ small, and regularity parameters are _commonly modest_, our method can be _guaranteed_ to achieve high accuracy" is overly vague. Given such an imprecise conclusion, the inclusion of Theorem 3.1 seems unnecessary.
- The reproducibility and preciseness of the dataset curation process seems to be limited. The conversion from video to static images results in nontrivial information loss during both training and inference, as not all objects produce sound continuously throughout the video in real life. Furthermore, the random selection of single frames from videos could potentially compromise both experimental accuracy and reproducibility.

**Questions:**

Some minor comments:
- Be consistent with the citation format in References.
- If speech and music are discarded with audio tagging model, how does the label distribution look like?

---

### Official Review · Reviewer_VR1V · 2024-10-31

**Soundness:** 2
**Presentation:** 3
**Contribution:** 2
**Rating:** 3
**Confidence:** 5

**Summary:**

This paper aims at generating sounds that accurately align with visual objects in complex scenes. The authors address the challenge of creating context-specific sounds for scenes with multiple sound sources, such as urban environments with varied ambient sounds (e.g., crowd noise, car engines, and wind). The model uses object-centric representations to link sounds to specific objects by enhancing a conditional latent diffusion model with dot-product attention. This approach allows the model to capture subtle audio details and facilitates user control over sound generation through object selection via segmentation masks.

**Strengths:**

1. The paper is well-structured and clearly presented, making it accessible and easy to follow.

2. Additionally, the authors provide a supplementary video that intuitively showcases the results, enhancing the clarity of the model's capabilities.

**Weaknesses:**

The paper faces several conceptual and methodological concerns.

1. The task itself may lack clear purpose: since a text prompt is required, using a large language model (LLM) to parse the prompt into object names could streamline the process without necessitating visual inputs. Given that this method builds on the text-to-audio AudioLDM model, conditioning audio generation directly on text prompts would likely be more effective and efficient than relying on images.

2. The theoretical analysis presented lacks practical evaluation. While the authors posit that the soft attention mechanism could be theoretically replaced with segmentation masks, no experiments are conducted to assess the impact of this substitution. Comparative experiments examining the performance of soft attention versus segmentation masks would strengthen this theoretical claim.

3. The proposed method also offers limited novelty, as it primarily integrates existing models with minor modifications.

4. The results raise questions about consistency. In Table 1, AudioLDM 1 is the weakest baseline, while Table 2’s "frozen diffusion" configuration—seemingly equivalent to AudioLDM 1—shows a sharp discrepancy in performance. Given that the method builds on AudioLDM, it is unclear if the proposed attention mechanism alone is responsible for transforming the baseline into a top-performing model, which may challenge the credibility of these results.

**Questions:**

Please refer to the weaknesses section.

---

### Official Review · Reviewer_HdH7 · 2024-11-03

**Soundness:** 3
**Presentation:** 3
**Contribution:** 3
**Rating:** 6
**Confidence:** 4

**Summary:**

This paper tackles a task of object-aware sound generation and proposes conditional diffusion model with object-centric representations by exploiting image-text attentions during training, and segmentation masks during test time. Authors validate the proposed model with both quantitative and qualitative evaluations and it shows the model outperforms baselines. Additionally, ablation study was conducted to demonstrate the effectiveness of the proposed components. Finally, the paper also presents theoretical analysis why the proposed object-grounding mechanism is equivalent to segmentation masks and thus segmentation masks may be used during inference.

**Strengths:**

- The idea of using object-centric representation for object-aware sound generation is interesting and makes sense.
- The experimental results are promising. Both quantitative and qualitative evaluation demonstrate the effectiveness of the proposed approach.
- Presented theoretical and ablation analyses are nice.
- In general, writing is easy to follow.

**Weaknesses:**

- Object-aware sound generation is not new. That is, there have been prior work using similar ideas for sound generation, e.g., [1] and [2]. I think the novelty is from explicitly modeling object-centric representation by using image-text attentions. But its novelty is somewhat incremental as this mechanism is not new, either.
- The baselines are not for object-aware sound generation, so it seems natural that the proposed model outperforms the baselines. I think it would be beneficial to compare the proposed model with other object-aware models for fairer comparison.
- Even though the authors present a theoretical analysis why the proposed object-grounding mechanism (i.e., image-text attention) is equivalent to segmentation masks and thus segmentation masks may be used during inference, it is still not clear why they are differently used in training and test time. e.g., can we use segmentation masks during training?




References

[1] Zhao et al., The sound of pixels, ECCV 2018

[2] Li et al., Cyclic Learning for Binaural Audio Generation and Localization, CVPR 2024

**Questions:**

- Using text-image attention at test time in ablation study, does it mean that it has access to text during inference? Or, was the text description generated with image captioning model such as BLIP?
- How about using segmentation masks during training instead of text-image attention since presumably they are equivalent?
- How about using both image and text for object-grounding mechanism, e.g., image-text-audio attention?

---

### Official Review · Reviewer_uetq · 2024-11-03

**Soundness:** 3
**Presentation:** 3
**Contribution:** 3
**Rating:** 6
**Confidence:** 3

**Summary:**

In this work, the authors propose an object-aware sound generation model that generates sounds aligned with visual objects in the scene. This approach overcomes the limitation of forgotten or underrepresented sound events in complex scenes. Results show that the framework can generate more complete and contextually relevant sounds and surpasses baselines.

**Strengths:**

1. The framework seems simple yet effective. By combining existing encoder and decoder blocks and fine-tuning the diffusion weights, the framework successfully grounds sound generation in object-centric representations. Results demonstrate the effectiveness of the framework.
2. The paper is well-written and easy to understand.

**Weaknesses:**

1. No ground-truth spectrums are provided in Figure 3, making it hard to directly compare the performance of different approaches.
2. The authors only provide the overall performance on the whole test set without breakup. The distribution of the test set is not clearly illustrated. The authors are encouraged to provide a more detailed analysis of their model's performance. Please refer to the 'Questions' part below.

**Questions:**

1. It is hard to directly judge the quality of generated sounds from current qualitative results in Figure 3, as no ground-truth is provided. Could you add ground-truth spectrums so that the readers can better compare the performance of different methods?
2. Could you clarify the distribution of the test set? Specifically, is it composed of single-source or multi-source audio samples, and what is the category distribution of the test set? Additionally, I recommend considering a division of the test set into subsets of varying difficulty or category distribution (e.g., human, animal, etc.). This could offer a more detailed evaluation of the framework's performance across different scenes.
3. The current image-text fusion method is relatively simple as only a cross-attention mechanism is adopted. Could you try different fusion mechanisms?

---

### Author Response · Authors · 2024-11-15
**General Response**

We thank the reviewer for the valuable comments and time.

**Clarify the dataset (uetq, TZCS)**

We first group the dataset into top-8 categories, i.e., Vehicle, Animal, Environment, Impact, Engine, Tool, Human, and Other, derived from AudioSet annotations. We then uniformly sample 48 hours across these categories for the test set, with the remaining used for training. As most clips contain multiple sound sources, we randomly select 100 examples from the test set to assess our model's ability to generate object-specific sounds through human evaluation. For 50 of these samples, we manually create object masks by splitting each caption into object snippets and then randomly selecting one to guide SAM in generating the mask.

**Other fusion mechanisms (uetq)**

We experimented with other fusion mechanisms, including multi-head attention fusion (Table 2(ii)) and additive attention fusion (Table 2(iv)) in our ablation study. However, these methods did not perform as well as dot-product attention when replacing segmentation masks at test time.

**Other object-aware models (HdH7)**

The suggested models [1, 2] are not compatible with our setting, where [1] focuses on audio separation and [2] on audio spatialization, both requiring input audio and not dedicated to sound generation. However, we adapted them for comparison. For [1], we first retrieve audio based on a text prompt using CLAP, then use the separation model to isolate the object-specific audio from the pixel of the visual object (Retrieve & Separate). As shown in the table below, our method outperforms this method across different objective metrics:

| Method    | ACC \(↑) \) | FAD \(↓\) | KL \(↓\) | IS \(↑\) | AVC \(↑\) |
|-----------|------------------------|----------------------|-----------------------|----------------------|----------------------|
| Retrieve & Separate    | 0.276                   | 4.051                | 1.572                  | 1.550                | 0.764                |
| **Ours**  | **0.859**               | **1.271**            | **0.517**              | **2.102**         | **0.891**                |

[2] is not open source, but we have already implemented audio-image attention similar to their method. The results are shown in Table 2(iii) of our paper, and their performance is not as good as ours. We will include this reference in a revision.

**Use segmentation masks during training (HdH7)**

Thanks for your suggestion. We trained a model using segmentation masks at both training and test time. As shown in the table below, this method degrades performance. We hypothesize that masking entire object regions imposes an overly rigid prior, as sound is typically emitted from specific parts (e.g., a dog's head rather than its tail). From a probabilistic perspective, hard masks sample from the ground truth distribution with high variance, whereas soft attention approximates this distribution with lower variance, aided by strong CLIP and CLAP embeddings. This allows the model to focus on sound-relevant regions while maintaining audio accuracy with uniformly weighted masks at test time.

**Clarify text-image attention at test time (HdH7)**

Yes, this model uses text at test time in this setting. We evaluate metrics using a held-out test set from the Sound-VECaps dataset, which includes text annotations.

**Clarify text-based over image-based methods (VR1V)**

We are unsure what you are referring to, as we do not argue that image-based methods are superior to text-based counterparts. Rather, we note that text-based methods may struggle to capture all desired audio elements, potentially missing some sounds due to differences in event weighting in the latent space [3]. To address this issue, we provide a complementary method that allows users to select which objects produce sound. Since segmentation masks assign similar weights across selected objects without prioritizing anyone, even subtle sound events can be accurately captured. Quantitative comparisons (especially ACC) in Table 1 demonstrate that our method generates more precise audio than text-based methods.

**Soft attention vs. segmentation masks (VR1V)**

We have conducted these experiments in Table 2(v) of our paper, comparing performance with and without replacing text-image attention with segmentation masks at test time. In Lines 460--464 of our paper, we show that text-image attention achieves performance on par with the segmentation mask approach (ours). This suggests that both methods provide similar levels of spatial and semantic guidance for audio generation.

---

> ### Author Response · Authors · 2024-11-15
>
> **Limited novelty (VR1V)**
>
> Our major contribution is  *not* a new network architecture or algorithm. Instead, it is identifying a natural correlation between objects and sound for audio generation. We are the first to learn audio generation from objects within a scene. That said, our model contributions *are also non-trivial*: we extend text-to-audio methods like AudioLDM to discover object-sound correlations for audio generation. We also present a novel application where users can select which objects produce sound, enabling interactive audio generation.
>
> **Results inconsistency in AudioLDM (VR1V)**
>
> We are puzzled by this concern, as these two models are quite different. Frozen Diffusion incorporates image conditioning, while AudioLDM 1 only supports text conditioning. Therefore, performance comparisons across variants may not be appropriate.
>
> **Clarify theoretical analysis (TZCS)**
>
> We have defined the value metric in Line 258 of our paper to measure the performance gap between our trained model and the best model in its class. The regularity parameters $L_v, L_f, B_v$ are standard in the learning theory literature [4, 5] and can be bounded with guarantees [6, 7]. In Line 283, we have explained the formal theorem stated earlier, clarifying how the terms in the bound relate to practice.
>
> **Reproducibility of dataset curation (TZCS)**
>
> As mentioned in the Reproducibility Statement of our paper, we have provided detailed information in multiple sections of this paper and the appendix. We will release code, dataset, and models upon acceptance.
>
> **Effect of the random selection of single frames from videos (TZCS)**
>
> We agree that randomly selecting single frames from videos may potentially affect performance. However, we have conducted a three-step data refinement process to ensure that the data contains continuous sound throughout the video, as detailed in Appendix A.2 (Lines 884--947) of our paper. Additionally, we have provided quantitative comparisons, as well as a results video, to demonstrate that our model successfully generates sound under this setting and can generalize to in-the-wild visual data.
>
> **Clarify the test set (TZCS)**
>
> We first group the dataset into top-8 categories, i.e., Vehicle, Animal, Environment, Impact, Engine, Tool, Human, and Other, derived from AudioSet annotations. We then uniformly sample 48 hours across these categories for the test set, with the remaining used for training. As most clips contain multiple sound sources, we randomly select 100 examples from the test set to assess our model's ability to generate object-specific sounds through human evaluation. For 50 of these samples, we manually create object masks by splitting each caption into object snippets and then randomly selecting one to guide SAM in generating the mask.
>
> **Figure 3, compositional term, and reference format (uetq, TZCS)**
>
> Thanks for your suggestion. We will correct them in the revised version.
>
> **References**
>
> [1] Zhao et al., The sound of pixels, ECCV 2018.
>
> [2] Li et al., Cyclic Learning for Binaural Audio Generation and Localization, CVPR 2024.
>
> [3] Wu et al. Audio-text models do not yet leverage natural language, ICASSP 2023.
>
> [4] Anthony et al. Neural Network Learning. Cambridge: Cambridge University Press, 1999.
>
> [5] Neyshabur et al. Norm-based capacity control in neural networks. COLT, 2018.
>
> [6] Tsuzuku et al. Lipschitz-margin training: Scalable certification of perturbation invariance for deep neural networks. NeurIPS, 2018.
>
> [7] Combettes et al. Lipschitz certificates for layered network structures driven by averaged activation operators. Journal on Mathematics of Data Science, 2020.

---

### Note · Authors · 2024-11-15

I have read and agree with the venue's withdrawal policy on behalf of myself and my co-authors.